# PRODIGY: Enabling In-context Learning Over Graphs

**Qian Huang**[1]*
qhwang@cs.stanford.edu

**Hongyu Ren**[1]*
hyren@cs.stanford.edu

**Peng Chen**[1]
pengc@stanford.edu

**Gregor Kržmanc**[2]
gregor.krzmanc@ijs.si

**Daniel Zeng**[1]
dzeng@cs.stanford.edu

**Percy Liang**[1]
pliang@cs.stanford.edu

**Jure Leskovec**[1]
jure@cs.stanford.edu

[1] Stanford University
[2] University of Ljubljana

## Abstract

In-context learning is the ability of a pretrained model to adapt to novel and diverse downstream tasks by conditioning on prompt examples, without optimizing any parameters. While large language models have demonstrated this ability, how in-context learning could be performed over graphs is unexplored. In this paper, we develop **Pr**etraining **O**ver **D**iverse **I**n-Context **G**raph S**y**stems (PRODIGY), the first pretraining framework that enables in-context learning over graphs. The key idea of our framework is to formulate in-context learning over graphs with a novel *prompt graph* representation, which connects prompt examples and queries. We then propose a graph neural network architecture over the prompt graph and a corresponding family of in-context pretraining objectives. With PRODIGY, the pretrained model can directly perform novel downstream classification tasks on unseen graphs via in-context learning. We provide empirical evidence of the effectiveness of our framework by showcasing its strong in-context learning performance on tasks involving citation networks and knowledge graphs. Our approach outperforms the in-context learning accuracy of contrastive pretraining baselines with hard-coded adaptation by 18% on average across all setups. Moreover, it also outperforms standard finetuning with limited data by 33% on average with in-context learning.

## 1 Introduction

In-context learning is a novel and one of the most intriguing capabilities of language models [1]. It refers to the capability of a pretrained model to perform novel and diverse tasks directly at the prediction time when prompted with just a few examples, without the need to update the model weights. For example, a person may describe the new task (*e.g.*, question answering, machine translation, or code generation) using natural language and demonstrate it to the language model with several prompt examples. The language model then directly without any model training or finetunning performs the task.

However, how to enable in-context learning for diverse graph machine learning tasks, such as identifying misinformation spreader in social networks [14] and product suggestions across online e-commerce websites [21], still remain unexplored and challenging. An in-context learner for graphs should be able to solve novel tasks on novel graphs. For example, give music product

---

*indicates equal contribution.

37th Conference on Neural Information Processing Systems (NeurIPS 2023).

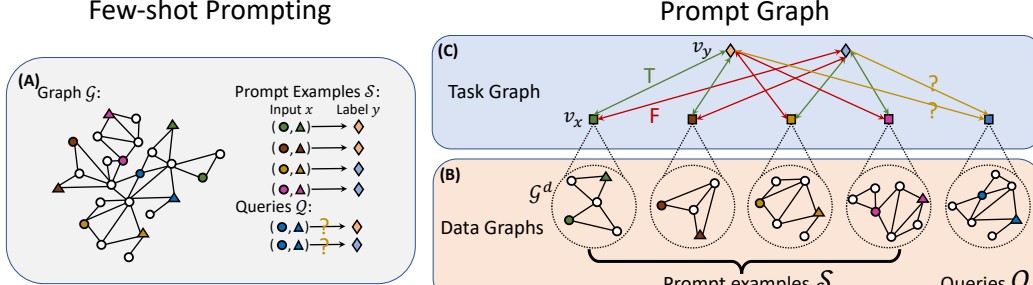

Figure 1: In-context few-shot prompting over graphs with prompt graph for *edge classification* in PRODIGY. (A) Given the source graph $\mathcal{G}$, we provide prompt examples $\mathcal{S}$ that consist of the input head/tail nodes and their labels, as well as the queries. (B) For each datapoint from both prompt examples and the queries, we first construct its data graph $\mathcal{G}^D$ by retrieving context from the source graph $\mathcal{G}$. (C) Then we create a task graph to capture the connection between each datapoint and each label, which includes a data node $v_x$ for each datapoint and a label node $v_y$ for each label in $\mathcal{Y}$. Each pair of data and label nodes are connected with edge attributes corresponding to their binary labels.

recommendations on Spotify when being trained on Amazon purchasing graph. The first challenge here is how to formulate and represent node-, edge- and graph-level tasks over graphs with a unified task representation that allows the model to solve diverse tasks without the need for retraining or parameter tuning. In other words, the key challenge is: what is an analog of natural language prompting for graph machine learning tasks? The second challenge is how to design model architecture and pretraining objectives that enable models to achieve in-context learning capability across diverse tasks and diverse graphs in the unified task representation. Existing graph pretraining methods [7, 24, 8, 13] only aim to learn a good graph encoder and require fine-tuning to adapt to different tasks, while existing meta-learning methods over graphs [19, 9, 3, 17, 25] only aim to generalize across different tasks within the same graph. On the other hand, achieving in-context learning requires tackling the more difficult setting of generalizing across the graphs *and* tasks without finetuning.

Here we present a general approach for solving these two challenges for classification tasks on graphs: (1) *prompt graph*, an in-context graph task representation, and (2) **Pr**etraining **O**ver **D**iverse **I**n-Context **G**raph S**y**stems (PRODIGY), a framework for pretraining an in-context learner over prompt graphs.

We propose *prompt graph* (Figure 1) to provide unified way to represent diverse node-, edge- and graph-level machine learning tasks. Prompt graph first contextualizes the input nodes/edges on which we make prediction (including both the prompt examples and the queries), then connects them with additional label nodes, such that the prompt examples are interconnected with queries. Such a unified representation allows us to specify diverse graph machine learning tasks to the same model regardless of the graph size.

PRODIGY then designs both model architecture and pretraining objectives with the prompt graph in-context task formulation, such that the model is pretrained to solve tasks across a wide range of tasks and graphs, and can continue to do so out-of-the-box. We design a graph architecture that utilizes graph neural networks to learn node/edge representations and an attention mechanism to communicate over prompt graph. Furthermore, we propose a family of in-context pretraining objectives over prompt graph. In particular, this includes a novel self-supervised pretraining task, *neighbor matching*, where we classify which neighborhood a node or edge belongs to.

We use PRODIGY framework to pretrain on citation networks (MAG240M [5]) and knowledge graphs (Wiki [22]). We then show that such model (without any retraining) provides strong performance on in-context paper category classification and knowledge graph completion tasks on novel graphs it was never trained on (arXiv, ConceptNet, FB15K-237, NELL) [6, 16, 23]. Specifically, PRODIGY improves upon contrastive pretraining baselines with hard-coded adaptation for in-context setup by 18% on average across all datasets and numbers of labels to classify among. Moreover, it also outperforms standard finetuning with limited data by 32.6% on average with in-context learning. It even outperforms the state-of-the-art few-shot learning methods trained on the testing downstream graph with pure in-context learning. Finally, we further demonstrate that our methods achieve increasingly higher performance with more examples in the prompt even beyond what it was pretrained with, which shows that the model really learns to learn from context.

## 2 In-context Learning over Graphs

In this work, we specifically focus on in-context learning for node and edge classification tasks on graphs with few-shot prompting, which are the forms of the most standard and important graph machine learning tasks. In this section, we introduce the concrete classification tasks over graphs and few-shot prompting over them with our in-context task representation prompt graph.

### 2.1 Classification Tasks over Graphs

We define a graph as $\mathcal{G} = (\mathcal{V}, \mathcal{E}, \mathcal{R})$, where $\mathcal{V}$, $\mathcal{E}$, $\mathcal{R}$ represent the set of nodes, edges and relations. An edge $e = (u, r, v) \in \mathcal{E}$ consists of a subject $u \in \mathcal{V}$, a relation $r \in \mathcal{R}$ and an object $v \in \mathcal{V}$.

Given a set of classes $\mathcal{Y}$, a standard classification task is predicting the labeling $y \in \mathcal{Y}$ of each input $x \in \mathcal{X}$. A node-level classification task is similar but each input is a single node in $\mathcal{G}$, *i.e.*, $\mathcal{X} = \mathcal{V}$, with the additional auxiliary information of the entire graph $\mathcal{G}$. For example, over a citation network consisting of authors and papers, a node-level classification task could be predicting the primary institution of each author. Similarly, an edge-level classification task is predicting the best labeling of potential edges formed by any pair of nodes, *i.e.*, $\mathcal{X} = \mathcal{V} \times \mathcal{V}$. A common special case is that the classes are the same as the relations $\mathcal{Y} = \mathcal{R}$, such as predicting the relation between entities over knowledge graphs. More generally, the same definitions can be extended to subgraph and graph-level classification tasks, where the input data $x$ may consist of more nodes and edges, and essentially represents a subgraph of $\mathcal{G}$.

Since we are interested in tasks of different types/levels, we design a unified formulation, where the space of the input $\mathcal{X}$ consists of graphs, *i.e.*, $x_i \in \mathcal{X}, x_i = (\mathcal{V}_i, \mathcal{E}_i, \mathcal{R}_i)$. For node classification task, $\mathcal{G}_i$ only consists of the input node that we aim to make predictions on, *i.e.*, $|\mathcal{V}_i| = 1$ and $|\mathcal{E}_i| = 0$; for edge classification task, it consists of (subject, object) pair, *i.e.*, $|\mathcal{V}_i| = 2$ and $|\mathcal{E}_i| = 0$.

### 2.2 Few-shot Prompting

Here we define in-context learning setup for classification tasks over graphs with few-shot prompting. For a $k$-shot prompt with a downstream $m$-way classification tasks with $|\mathcal{Y}| = m$ classes, we use a small number of input-label pairs $\mathcal{S} = \{(x_i, y_i)\}_{i=1}^{m \cdot k}$ as *prompt examples* of the task specification, such that there are $k$ input-label pairs with label $y$ for each $y \in \mathcal{Y}$. We also give the model a set of queries $\mathcal{Q} = \{x_i\}_{i=1}^{n}$ that we want to predict labels for.

We emphasize an important difference of classification tasks on graphs from language and other modalities. Namely, since all input datapoints are nodes/edges/subgraphs from the larger source graph $\mathcal{G}$, this graph contains critical information and provides contexts for the inputs, *e.g.*, the local neighborhood of the input node that we aim to predict. Hence, besides $\mathcal{S}$ and $\mathcal{Q}$, we also need to include the source graph $\mathcal{G}$ in the prompt.

Given the above information as the prompt, the pretrained model should be able to directly output the predicted labels for each datapoint in $\mathcal{Q}$ via in-context learning. Thus, how to formulate the information as a unified and efficient form of input poses a unique challenge and affects the model architecture. Below, we present our in-context task formulation prompt graph designed to do so.

### 2.3 Prompt Graph Representation

Inspired by [2], we propose prompt graph as a unified representation of a $k$-shot prompt over graphs for an $m$-way classification task (Figure 1). A prompt graph is composed of *data graphs* and a *task graph*:

**Data graph.** To construct a prompt graph, we first perform contextualization of each datapoint $x_i = (\mathcal{V}_i, \mathcal{E}_i, \mathcal{R}_i)$ in $\mathcal{S}$ and $\mathcal{Q}$ in the source graph $\mathcal{G}$ to form data graphs. The goal is to gather more information about the $x_i$ from the source graph $\mathcal{G}$ without having to represent the entire source graph explicitly. There are many potential designs for contextualization, from explicitly retrieving subgraphs to implicitly using embedding-based methods. Here we construct data graph $\mathcal{G}_i^{\mathrm{D}}$ by sampling $k$-hop neighborhood of $\mathcal{V}_i$ in $\mathcal{G}$. In other words, $\mathcal{G}_i^{\mathrm{D}} = (\mathcal{V}_i^{\mathrm{D}}, \mathcal{E}_i^{\mathrm{D}}, \mathcal{R}_i^{\mathrm{D}}) \sim \bigoplus_{i=0}^{k} \texttt{Neighbor}(\mathcal{V}_i, \mathcal{G}, i)$, where $\mathcal{V}_i \subseteq \mathcal{V}_i^{\mathrm{D}} \subseteq \mathcal{V}$, $\mathcal{E}_i \subseteq \mathcal{E}_i^{\mathrm{D}} \subseteq \mathcal{E}$, $\mathcal{R}_i \subseteq \mathcal{R}_i^{\mathrm{D}} \subseteq \mathcal{R}$, and $\texttt{Neighbor}$ is a function that returns the exact $i$-hop neighbors of each node in $\mathcal{V}_i$. With this data graph $\mathcal{G}_i^{\mathrm{D}}$, we call the node set that corresponds to the nodes in $\mathcal{V}_i$ before contextualization *input node set*, *e.g.*, the target node to classify in node classification task and the pair of nodes in link prediction task.

**Task graph.** After contextualizing each datapoint to a data graph $\mathcal{G}^\mathrm{D}$, we then construct task graph $\mathcal{G}^\mathrm{T}$ to better capture the connection and relationship among the inputs and the labels. For each data graph $\mathcal{G}^\mathrm{D}_i$ from the previous stage, we have a *data node* $v_{x_i}$ that represents each input; for each label, we have a *label node* $v_{y_i}$. So overall, a task graph contains $m \cdot k + n$ data nodes ($m \cdot k$ prompt examples and $n$ queries) and $m$ label nodes, as shown in Figure 1.

Now we add edges between the data nodes and the label nodes: For the query set, since we do not know the labels of each graph, we add single directional edges from all label nodes to each datapoint in the query set, *i.e.*, each query data node $v_{x_i}$ will be connected to all the label nodes as shown by the yellow edges in Figure 1; For the prompt examples, we connect each data node to all the label nodes, where the edge with the true labels is marked as T while the others are marked as F, as shown by the green and red edges in Figure 1 respectively.

Together we propose the prompt graph that consists of both data graphs and a task graph. Prompt graph effectively captures the relationship between input data $x_i$ and the label $y_i$ through the context captured in data graph $\mathcal{G}^\mathrm{D}_i$ and the data node $v_{x_i}$ and the label node $v_{y_i}$ in the task graph $\mathcal{G}^\mathrm{T}$. It is also possible to extend prompt graph to non-classification tasks and free-form text prompting. For example, for numerical regression (e.g. molecular energy prediction) and other free-form generation tasks (e.g. text generation), one can extend our task graph to contain vector values on the edges to represent $y_i$. Then different label nodes would represent different prediction tasks. To support more general forms of prompting, one can include additional task information and instructions in the feature of label nodes, and additional description paired with each datapoint in the global feature in data graph.

## 3 Pretraining to Enable In-context Learning

So far given a few-shot prompt for a classification task over graphs, we have defined a prompt graph representation for it that captures relationships between the prompt examples, queries, and labels. Now we need to design a pretraining strategy that can pretrain a generalizable model capable of in-context learning. We assume access to a pretraining graph $\mathcal{G}_{\texttt{pretrain}}$ that is independent of the source graph $\mathcal{G}$ for the downstream task.

In this section, we introduce PRODIGY, a general pretraining framework over $\mathcal{G}_{\texttt{pretrain}}$ that is designed specifically for enabling in-context learning over downstream classification tasks without any additional finetuning steps on arbitrary graphs. Our framework PRODIGY has two main components: model architecture over prompt graph and in-context pretraining objectives.

### 3.1 Message Passing Architecture over prompt graph

Next we introduce our model architecture over the prompt graph consisting of two submodules:

**Data graph Message Passing.** First, we apply a message passing GNN module $M_\mathrm{D}$ that learns node representation $E$ for nodes in each $\mathcal{G}^\mathrm{D}$.

$$E \in \mathcal{R}^{|\mathcal{V}^\mathrm{D}| \times d} = M_\mathrm{D}(\mathcal{G}^\mathrm{D}) \tag{1}$$

where $d$ is the embedding dimension. $M_\mathrm{D}$ can be implemented in multiple ways, such as using Graph Convolutional Network (GCN) or Graph Attention Networks (GAT) [11, 18].

To read out a single embedding $G_i$ for each data graph, we perform another aggregation step to pool node embeddings. For node classification tasks, we take the updated node representation of the single input node that we aim to predict, *i.e.*:

$$G_i = E_{\mathcal{V}_i} \tag{2}$$

For link prediction tasks, we concatenate the pair of nodes, which we want to predict a link between, as well as a max pooling over all node representations following [10] with an additional linear projection layer at the end to convert the embedding size back to $d$.

$$G_i = W^T(E_{v_1 \in \mathcal{V}_i} || E_{v_2 \in \mathcal{V}_i} || \max(E_i)) + b, \tag{3}$$

where $||$ represents concatenation, $W \in \mathcal{R}^{3d \times d}$ is a learnable weight matrix and $b$ is the learnable bias.

**Task graph Message Passing.** Note in the previous step there is no communication between different datapoints in $\mathcal{S}$ and $\mathcal{Q}$. Now we would like to communicate between them via message passing over

the task graph $\mathcal{G}^\mathrm{T}$. We apply another GNN $M_\mathrm{T}$ on the task graph to obtain updated representation of data nodes and label nodes.

$$H = M_\mathrm{T}(\mathcal{G}^\mathrm{T}) \tag{4}$$

where H is the obtained embedding per node. The initial embedding of data node $v_{x_i}$ is $G_i$ and the embedding of label node $v_{y_i}$ can either be initialized with random Gaussian or additional information available about the labels. Each edge also has two binary features $e_{ij}$ that indicate 1) whether the edge comes from an example or a query, and 2) the edge type of T or F. For $M_\mathrm{T}$, we use an attention-based GNN, where each node performs attention to other nodes at each layer. See the architecture detail in the appendix C.

The goal of this step is to learn a better representation of the label nodes using the support examples and propagate label information back to the support and query graph representation for a more task-specific graph representation.

**Prediction Read Out.** Finally, we readout the classification logits $O_i$ by taking cosine similarity between each pair of query graph representation and label representation, as in contrastive learning:

$$O_i = [\texttt{cosine\_similarity}(H_{x_i}, H_y), \forall y \in \mathcal{Y}] \tag{5}$$

Note that we could perform the two message passing steps for multiple rounds to have more communication between $x_i$ and learn a better representation. One key insight is that different in-context prompt examples share information through the label nodes, which can be seen as an information bottleneck.

## 3.2 In-context Pretraining Objectives

In order to pretrain the model for solving the downstream graph tasks in-context, we propose a set of in-context pretraining objectives. The goal is to pretrain the graph model using a large pretraining graph $\mathcal{G}_\texttt{pretrain}$ independent of the downstream task graph, such that the model can directly be applied on downstream tasks with in-context learning.

Our main design principle is that we formulate each pretraining objective in an in-context learning way. Most previous graph pretraining objectives only pretrain a shared graph encoder to perform various tasks with task-specific heads, so they require finetuning for another task-specific head over each downstream task. In contrast, we explicitly construct in-context pretraining tasks in prompt graph form and pretrain the model to solve diverse tasks in-context with the same set of weights, such that it can perform in-context learning directly over downstream tasks.

Below, we detail our proposed family of in-context pretraining objectives in terms of three components: 1) pretraining task generation, including few-shot prompt (*i.e.* Figure 1(A)) and corresponding labels, 2) converting generated few-shot prompt to prompt graph format (*i.e.* Figure 1(B,C)) with augmentation, and 3) pretraining loss over the generated prompt graph.

### 3.2.1 Pretraining Task Generation

We propose two methods to generate pretraining tasks from the pretraining graph $\mathcal{G}_\texttt{pretrain}$ in the form of few-shot prompts: *neighbor matching* and *multi-task*.

**Neighbor Matching.** Given the pretraining graph, we construct self-supervised in-context pretraining tasks with the goal of classifying which local neighborhood a node belongs to, where each local neighborhood is defined by the example nodes belonging to that neighborhood. Intuitively, we sample multiple subgraphs from the pretraining graph $\mathcal{G}_\texttt{pretrain}$ as the local neighborhoods, and we say a node belongs to a local neighborhood if it is in the sampled subgraph.

Formally, we denote $\texttt{NM}_{k,m}$ as a sampler that generates $m$-way neighbor matching tasks, where each includes a $k$-shot prompt $(\mathcal{G}_\texttt{pretrain}, \mathcal{S}_\mathrm{NM}, \mathcal{Q}_\mathrm{NM})$ (see subsection 2.2 and Figure 1(A)) and the labels of the queries. For simplicity of the notation, we will include the labels in $\mathcal{Q}_\mathrm{NM}$ as paired with the inputs:

$$(\mathcal{G}_\texttt{pretrain}, \mathcal{S}_\mathrm{NM}, \mathcal{Q}_\mathrm{NM}) \sim \texttt{NM}_{k,m}(\mathcal{G}_\texttt{pretrain}) \tag{6}$$

To generate these, we first sample $m$ nodes from the pretraining graph $\mathcal{G}_\texttt{pretrain}$, where each of the sampled node corresponds to one class.

$$\mathcal{C} = \{c_i\}_{i=1}^m \quad c_i \sim Uniform(\mathcal{V}_\texttt{pretrain}) \tag{7}$$

For each sampled node/class $c_i$, we sample $k$ different nodes from its exact $l$-hop neighbors. These $k$ nodes serve as examples of label $c_i$. We also sample additional $\lceil \frac{n}{m} \rceil$ nodes similarly for each label $c_i$ to form the query set. Formally,

$$N_i = \texttt{Neighbor}(c_i, \mathcal{G}_{\text{pretrain}}, l) \tag{8}$$

$$\mathcal{S}_i = \{(x_j, y_j = c_i)\}_{j=1}^{k} \quad x_j \sim Uniform(N_i) \tag{9}$$

$$\mathcal{Q}_i = \{(x_j, y_j = c_i)\}_{j=1}^{\lceil \frac{n}{m} \rceil} \quad x_j \sim Uniform(N_i) \tag{10}$$

In such a way, we constructed a neighbor matching pretraining task sample in the format of a few-shot prompt $(\mathcal{G}_{\text{pretrain}}, \mathcal{S}_{\text{NM}} = \bigcup \mathcal{S}_i, \mathcal{Q}_{\text{NM}} = \bigcup \mathcal{Q}_i)$.

The neighbor matching task generation process outlined above is specifically applicable when the downstream tasks are also node classification. When the downstream task is link prediction, we may adapt the above neighbor matching tasks to over edges correspondingly. Specifically, we can expand each sampled input node $x_i$ to an edge by randomly sampling an edge that contains $x_i$. Then, instead of classifying to which neighborhood a node in the query set belongs, now the neighbor matching task is to classify to which neighborhood an edge in the query set belongs.

**Multi-task.** When the pretraining graphs have node or edge-level labeling $f(x_i) = y_i \in \mathcal{Y}$ for some $x_i \in \mathcal{V}_{\text{pretrain}}$ or $\mathcal{E}_{\text{pretrain}}$, we can further leverage this signal to perform supervised pretraining. Similar to neighbor matching, the key is to construct such supervised pretraining tasks in the format of few-shot prompts and corresponding labels.

$$(\mathcal{G}_{\text{pretrain}}, \mathcal{S}_{\text{MT}}, \mathcal{Q}_{\text{MT}}) \sim \texttt{MT}_{k,m}(\mathcal{G}_{\text{pretrain}}, f) \tag{11}$$

For node classification tasks, we first sample $m$ labels from the whole label set. Then, for each label, we directly sample $k$ nodes as support examples and $\lceil \frac{n}{m} \rceil$ nodes with labels in this set as query examples.

$$\mathcal{C} = \{c_i\}_{i=1}^{m} \quad c_i \sim Uniform(\mathcal{Y}) \tag{12}$$

$$\mathcal{S}_i = \{(x_j, y_j = c_i)\}_{j=1}^{k} \quad x_j \sim Uniform(\{x_i | f(x_i) = c_i\}) \tag{13}$$

$$\mathcal{Q}_i = \{(x_j, y_j = c_i)\}_{j=1}^{\lceil \frac{n}{m} \rceil} \quad x_j \sim Uniform(\{x_i | f(x_i) = c_i\}) \tag{14}$$

We then construct a task with the few-shot prompt as $(\mathcal{G}_{\text{pretrain}}, \mathcal{S}_{\text{MT}} = \bigcup \mathcal{S}_i, \mathcal{Q}_{\text{MT}} = \bigcup \mathcal{Q}_i)$. For link prediction, we directly use the edge type function as $f$, i.e. $f((v_1, v_2)) = r \iff (v_1, r, v_2) \in \mathcal{E}$. With this $f$, we may directly sample $m$ edge types and construct pretraining tasks in a similar way as above.

The benefit of such a supervised pretraining objective is that it could directly resemble the format of downstream tasks, compared with neighbor matching objective, which may only serve as a surrogate. However, it requires extra labels if $f$ is not part of $\mathcal{G}_{\text{pretrain}}$, e.g. node classification labels that may not exist for some pretraining graphs.

### 3.2.2 Prompt graph generation with augmentation

After we obtained the few-shot prompts and labels for either of the two tasks (NeighborMatching and multi-task), we need to construct the prompt graph for pretraining. In addition to the standard construction process described in subsection 2.3, we add an additional augmentation step to augment

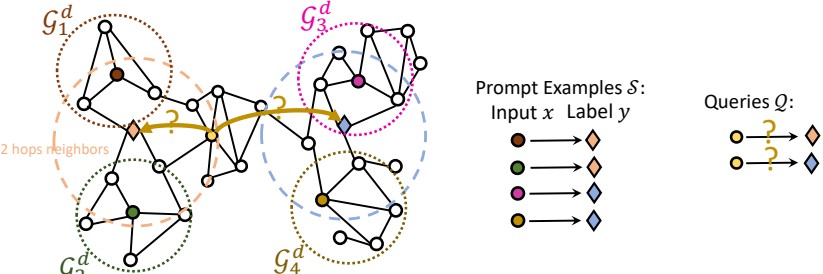

Figure 2: Neighbor matching pretraining task generation. The key idea of neighbor matching is to classify whether a node/edge is in the local neighborhood of a set of sampled nodes (as labels). Given a set of sampled nodes and their two-hop neighbors as prompt examples, we aim to classify whether a query node is also a two-hop neighbor of each sampled node.

the data graphs as inspired by Contrastive Learning. The key insight is to corrupt data graph such that the pretrained model learns representation invariant to various corruptions.

Here we demonstrate how we adopt graph augmentation techniques during the construction of prompt graph from a few-shot prompt generated from $\mathcal{G}_{\texttt{pretrain}}$. We first still sample the $k$-hop neighbor sub-graph of each sample $\mathcal{G}_i$ in the prompt examples and queries: $\mathcal{G}_i^{\texttt{D}} \sim \bigoplus_{j=1}^{k} \texttt{Neighbor}(\mathcal{G}_i, \mathcal{G}_{\texttt{pretrain}}, j)$. Then we adopt the following two augmentation techniques to create augmented data graph $\mathcal{G}_i^{aug}$, including (1) node dropping, and (2) node feature masking [24]. For node dropping, we randomly drop nodes from the $k$-hop neighbor subgraph and take the remaining graph as $\mathcal{G}_i^{aug} = \texttt{DropNode}(\mathcal{G}_i^{\texttt{D}})$. For node feature masking, we randomly mask the feature of a subset of nodes with value zero to create $\mathcal{G}_i^{aug} = \texttt{MaskNode}(\mathcal{G}_i^{\texttt{D}})$. With the augmented data graphs for each datapoint in the prompt examples and the queries, we may accordingly construct the task graph $\mathcal{G}^{\texttt{T}}$ by creating a data node $v_{x_i}$ for each augmented data graphs and the label node $v_{y_i}$ as introduced in subsection 2.3. Combining data graphs with task graph, we obtain the prompt graph formulation with augmentation for the few-shot prompt.

### 3.2.3 Pretraining Loss

Finally, we pretrain the model with the cross-entropy objectives over generated prompt graphs:

$$(\mathcal{G}_{\texttt{pretrain}}, \mathcal{S}_{\text{NM}}, \mathcal{Q}_{\text{NM}}) \sim \texttt{NM}_{k,m}(\mathcal{G}_{\texttt{pretrain}}) \tag{15}$$

$$(\mathcal{G}_{\texttt{pretrain}}, \mathcal{S}_{\text{MT}}, \mathcal{Q}_{\text{MT}}) \sim \texttt{MT}_{k,m}(\mathcal{G}_{\texttt{pretrain}}, f) \tag{16}$$

$$\mathcal{L} = \underset{x_i \in \mathcal{Q}_{\text{NM}}}{\mathbb{E}} \texttt{CE}(O_{\text{NM},i}, y_{\text{NM},i}) + \underset{x_i \in \mathcal{Q}_{\text{MT}}}{\mathbb{E}} \texttt{CE}(O_{\text{MT},i}, y_{\text{MT},i}) \tag{17}$$

where $O_{\text{NM},i}$ is the logits produced by our model over input of $\mathcal{G}_i^{aug}$ and $\mathcal{G}^{\texttt{T}}$ produced from $\mathcal{Q}_{\text{NM}}$, as described in subsection 3.1; $y_{\text{NM},i}$ is the corresponding label of $x_i$ in $\mathcal{Q}_{\text{NM}}$; Similar for MT terms.

## 4 Experiments

### 4.1 Experimental Setup

**Datasets.** For pretraining, we use two datasets: `MAG240M` [5], a large scale citation network with 122 million nodes and 1.3 billion edges; and `Wiki`, a knowledge graph (KG) constructed from Wikipedia [22] with 4.8 million nodes and 5.9 million edges. After the model is pretrained we evaluate its in-context learning capability on diverse classification tasks over 4 graphs: `arXiv` [6], `ConceptNet` [16], `FB15K-237` [23], `NELL` [23]. We use subsets of knowledge graph datasets same as in [10, 23]. For arXiv, the downstream task is an $m$-ways node classification task that predicts the paper category. For knowledge graph datasets (`ConceptNet`, `FB15K-237`, `NELL`), the downstream task is an $m$-ways relation type classification task that predicts the relationship connecting the two input nodes.

**Evaluation.** We pretrain our model on `MAG240M` and `Wiki` and then we evaluate the in-context learning performance on differnt downstream datasets that belong to similar domain as the pretraining dataset (e.g., pretraining on `Wiki` and evaluating on `ConceptNet`, `FB15K-237`, and `NELL`). Each of the downstream classification datasets has its original train, validation, and test splits. To simulate the situation where there are a limited amount of labeled data in the downstream task, we randomly select 10 nodes (or edges) from the training split per way as the prompt examples with known labels. Then, we construct a $k$-shot prompt for test nodes (or edges) from the test split by randomly selecting $k$ examples per way from these available examples. This allows us to test the model's ability to learn in-context relationships and perform well on classification tasks with truly limited known labels. By default we use $k = 3$ shots in our experiments.

**Methods and Baselines.** We consider three versions of our proposed framework PRODIGY: 1) PG-NM, which uses neighbor matching task for pretraining; 2) PG-MT, which employs multi-task pretraining; and 3) full PRODIGY, which combines the previous two methods. To augment the data, we use DropNode and MaskNode augmentations with a probability of 0.5 per node for each method.

We consider three baselines for comparison: 1) NoPretrain, which uses a randomly-initialized model with the same architecture as our pretrained models; 2) Contrastive [24], which employs a standard contrastive learning method with the same augmentation as above and uses a hard-coded nearest neighbor algorithm to adapt to our in-context learning setting. Specifically, we classify the query by

Table 1: In-context learning accuracy (%) on arXiv paper category classification on 500 sampled test tasks with 3-shot prompts. PRODIGY was pretrained on MAG240M and is then applied in-context to arXiv, which has completely different structure and a different set of paper categories. PG-NM and PG-MT are ablations of PRODIGY.

| Classes | NoPretrain | Contrastive | PG-NM | PG-MT | PRODIGY | Finetune |
|---|---|---|---|---|---|---|
| 3 | $33.16_{\pm 0.30}$ | $65.08_{\pm 0.34}$ | $72.50_{\pm 0.35}$ | $65.64_{\pm 0.33}$ | $\mathbf{73.09}_{\pm \mathbf{0.36}}$ | $65.42_{\pm 5.53}$ |
| 5 | $18.33_{\pm 0.21}$ | $51.63_{\pm 0.29}$ | $61.21_{\pm 0.28}$ | $51.97_{\pm 0.27}$ | $\mathbf{61.52}_{\pm \mathbf{0.28}}$ | $53.49_{\pm 4.61}$ |
| 10 | $9.19_{\pm 0.11}$ | $36.78_{\pm 0.19}$ | $46.12_{\pm 0.19}$ | $37.23_{\pm 0.20}$ | $\mathbf{46.74}_{\pm \mathbf{0.20}}$ | $30.22_{\pm 3.77}$ |
| 20 | $4.72_{\pm 0.06}$ | $25.18_{\pm 0.11}$ | $33.71_{\pm 0.12}$ | $25.91_{\pm 0.12}$ | $\mathbf{34.41}_{\pm \mathbf{0.12}}$ | $17.68_{\pm 1.15}$ |
| 40 | $2.62_{\pm 0.02}$ | $17.02_{\pm 0.07}$ | $23.69_{\pm 0.06}$ | $17.19_{\pm 0.08}$ | $\mathbf{25.13}_{\pm \mathbf{0.07}}$ | $8.04_{\pm 3.00}$ |

Table 2: In-context learning accuracy (%) on ConceptNet, FB15K-237 and NELL (from top to bottom) on 500 sampled test tasks with 3-shot prompts. PRODIGY was pretrained on Wiki, which has completely different node and relation types from graphs it is then applied on in-context.

| Classes | NoPretrain | Contrastive | PG-NM | PG-MT | PRODIGY | Finetune |
|---|---|---|---|---|---|---|
| 4 | $30.4_{\pm 0.63}$ | $44.01_{\pm 0.61}$ | $46.94_{\pm 0.61}$ | $51.78_{\pm 0.63}$ | $\mathbf{53.97}_{\pm \mathbf{0.63}}$ | $53.85_{\pm 9.29}$ |
| 5 | $33.54_{\pm 0.61}$ | $81.35_{\pm 0.58}$ | $80.35_{\pm 0.57}$ | $89.15_{\pm 0.46}$ | $\mathbf{88.02}_{\pm \mathbf{0.48}}$ | $82.01_{\pm 12.83}$ |
| 10 | $20.0_{\pm 0.35}$ | $70.88_{\pm 0.48}$ | $71.68_{\pm 0.45}$ | $\underline{82.26}_{\pm 0.40}$ | $\mathbf{81.1}_{\pm \mathbf{0.39}}$ | $71.97_{\pm 6.16}$ |
| 20 | $9.2_{\pm 0.18}$ | $59.8_{\pm 0.35}$ | $59.9_{\pm 0.35}$ | $\underline{73.47}_{\pm 0.32}$ | $\mathbf{72.04}_{\pm \mathbf{0.33}}$ | $64.01_{\pm 4.66}$ |
| 40 | $2.5_{\pm 0.08}$ | $49.39_{\pm 0.23}$ | $46.82_{\pm 0.21}$ | $58.34_{\pm 0.22}$ | $\mathbf{59.58}_{\pm \mathbf{0.22}}$ | $57.27_{\pm 3.33}$ |
| 5 | $20.95_{\pm 0.52}$ | $83.38_{\pm 0.5}$ | $82.39_{\pm 0.53}$ | $85.26_{\pm 0.48}$ | $\mathbf{88.09}_{\pm \mathbf{0.43}}$ | $87.22_{\pm 12.75}$ |
| 10 | $11.0_{\pm 0.26}$ | $74.54_{\pm 0.46}$ | $75.14_{\pm 0.43}$ | $78.15_{\pm 0.41}$ | $\mathbf{82.47}_{\pm \mathbf{0.39}}$ | $71.90_{\pm 5.90}$ |
| 20 | $5.34_{\pm 0.13}$ | $65.68_{\pm 0.34}$ | $65.68_{\pm 0.34}$ | $68.38_{\pm 0.33}$ | $\mathbf{74.72}_{\pm \mathbf{0.31}}$ | $66.19_{\pm 8.46}$ |
| 40 | $2.5_{\pm 0.06}$ | $56.7_{\pm 0.23}$ | $54.91_{\pm 0.22}$ | $51.24_{\pm 0.25}$ | $\mathbf{60.04}_{\pm \mathbf{0.23}}$ | $55.06_{\pm 4.19}$ |

comparing its pretrained embedding against the average embedding of the example inputs of each class. 3) Finetune [7], which trains an additional linear classification head on top of the graph encoder pretrained with contrastive learning, following the standard practice.

## 4.2 In-Context Learning Results

We first evaluate the in-context learning capability for node classification and link prediction with various numbers of ways (i.e. number of classes to classify among). The results are presented in Table 1 and 2 for node classification and link prediction.

**Strong in-context learning performance.** The results demonstrate that our method PRODIGY consistently outperforms all other baselines in this setting. It achieves the highest average accuracy across all ways on `arXiv`, with an average improvement of 28.6% and up to 48% over the best baseline of Contrastive. Over KGs, PRODIGY also outperforms contrastive learning on average by 12.2%. PRODIGY also demonstrates similar-to-better performance compared to Finetune, which requires additional training on downstream tasks. On `arXiv`, we see an average improvement of 77.7% over all ways. This can be attributed to the diverse set of pretraining tasks incorporated in PRODIGY, which allows the model to avoid overfitting on specific tasks and learn in-context.

**Self-supervised pretraining PG-NM bridges different tasks.** In particular, we highlight that the pure self-supervised pretraining method PG-NM produces significantly higher in-context learning performance over `arXiv` than baselines, even though the model is pretrained on different tasks from the downstream task. This advantage can be further leveraged by pretraining on even larger-scale unlabeled datasets. On the other hand, PG-MT follows the supervised pretraining objective that directly resembles the format of downstream tasks. On KGs, this allows PG-MT to adapt better to downstream task even sometimes compared to the full PRODIGY ( marked by underlines), while PG-NM might have overfitted to the incorrect strategy of only identifying co-occurring nodes. Yet, PG-MT performs worse on `arXiv` potentially due to less diversity. The full PRODIGY, which ensembles the two, achieves more diversity than either single task and therefore achieves the best performance over both worlds.

**Outperforming meta-learning method trained on test graph.** Finally, we compare PG-NM in-context learning performance against state-of-the-art meta-learning method TENT [20] over the downstream test graph `arXiv`. We evaluate the average 3-ways classification tasks performance over only test labels, since TENT trains on train labels from `arXiv`. PG-NM achieves 69.07% over the 65.13% of TENT, even though PG-NM has never been trained on any paper category classification

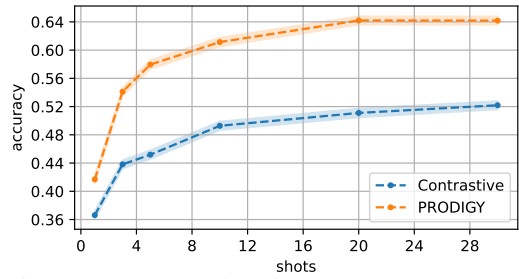

Figure 3: In-context learning accuracy on Concept-Net in a 4-ways setting wrt. the number of prompt examples (shots).

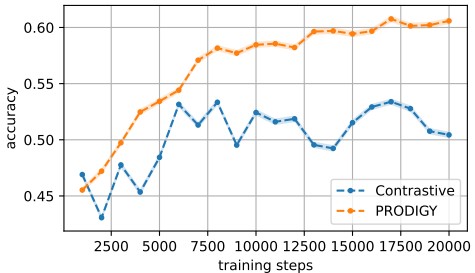

Figure 4: With more training steps and data on arXiv (5-ways), PRODIGY keeps improving while the baseline (Contrastive) saturates.

task during pretraining. This demonstrates the power of self-supervised pretraining over large amount of data compared to supervised meta-learning over the limited labeled data (train labels in `arXiv`).

### 4.3 Ablations

Aside from PG-NM and PG-MT, we also conduct ablation studies on various configurations of the self-supervised objective PG-NM as described in 3.2. See the full results in Appendix E and Table 4. Overall, the ablation results reveal that using all of the elements together results in the highest performance. Specifically, attribute prediction (see appendix A) has the greatest impact on PG-NM's performance, as its removal results in an average 7% drop across all ways, shown in the 'No-Attr' column.

### 4.4 Evaluation using different numbers of in-context examples

We investigate our method's ability to learn from the context by analyzing its performance as the number of prompt examples changes. Figure 3 shows the result on `ConceptNet`. See full results on other datasets in Appendix F. As the number of prompt examples increases, the margin of our proposed PG models over the baseline increases. This supports the hypothesis that the PRODIGY models can more effectively learn the unknown task by reasoning about the common characteristics of prompt examples.

### 4.5 Scaling with Data Size

Finally, we explore how the model scales with more pretraining data. The result on arXiv in a 5-ways setting is illustrated in Figure 4. It shows that the Contrastive baseline saturates quickly and its performance fluctuates as trained over more pretraining data. Instead, PRODIGY consistently shows an improvement in performance as more data is pretrained on, since the pretraining tasks are harder and more diverse.

## 5 Related Work

### 5.1 In-context Learning of Large Language Models

Pretrained large language models can make predictions for diverse downstream tasks directly by prompting with a few examples of the task or more generally any textual instructions. This ability is called in-context learning. Comparing to previous language encoder models like BERT [4], it drastically reduces the adaptation effort comparing to fine-tuning, and has demonstrated strong performance in a broad range of models and tasks. Our work extends this success similarly to graph data compared to the current pretrained graph encoders, such that a single pretrained model can be adapted to different classification tasks over different graphs without additional fine-tuning but only few-shot prompting.

### 5.2 Pretraining on Graphs

There are many existing works on pretraining over graphs[7, 24, 8, 13]. However, they all follow the general paradigm of learning a good graph encoder that can perform certain pretraining tasks, such as masked feature prediction [7] and paired graph classification [24]. To adapt to any downstream tasks, it then requires finetuning a classification head on top of the encoder with large amount of task specific data for each downstream task. In contrast, we explore pretraining methods for inducing general in-context learning ability, such that the pretrained model can be directly used for various downstream tasks with no gradient updates.

### 5.3 Meta Learning on Graphs

Another closely related line of works is meta-learning methods over graphs that aim to address standard few shot learning problems over graphs[19, 9, 3, 17, 25]. However, existing meta-learning methods are only designed and tested for generalizing across different tasks on the same graph: the methods are trained on a set of training tasks on a graph, then tested over a disjoint but similar set of test tasks over the same graph. They are shown to exhibit optimal performance only when trained on similar curated tasks [10]. Different from this, our work explicitly focuses on the in-context learning performance, i.e. model performance on graphs and tasks completely different from the pretraining without additional fine-tuning.

## 6 Conclusion

We introduce PRODIGY, the first framework that enables in-context learning on graphs. A model that is pretrained using PRODIGY can seamlessly execute a new classification task over new graphs represented by prompt graph. It markedly surpasses the performance of other baseline models with in-context learning, even those that employ finetuning, in both the node and edge classification tasks.

## Acknowledgments and Disclosure of Funding

We thank Camilo Ruiz and Michael Moor for discussions and for providing feedback on our manuscript. We also gratefully acknowledge the support of DARPA under Nos. HR00112190039 (TAMI), N660011924033 (MCS); ARO under Nos. W911NF-16-1-0342 (MURI), W911NF-16-1-0171 (DURIP); NSF under Nos. OAC-1835598 (CINES), OAC-1934578 (HDR), CCF-1918940 (Expeditions), NIH under No. 3U54HG010426-04S1 (HuBMAP), Stanford Data Science Initiative, Wu Tsai Neurosciences Institute, Amazon, Docomo, GSK, Hitachi, Intel, JPMorgan Chase, Juniper Networks, KDDI, NEC, and Toshiba. Hongyu Ren is supported by Masason Foundation PhD fellowship, Apple PhD fellowship and Baidu PhD scholarship. Qian Huang is supported by Open Philanthropy AI fellowship.

The content is solely the responsibility of the authors and does not necessarily represent the official views of the funding entities.

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

# A  Attribute Prediction Loss

For each augmented DataGraph $\mathcal{G}^{aug}$, the certain node features $F_v$ are masked during MaskNode augmentation. Therefore, we can reconstruct them using the learned embedding $E_v$ with a MLP and train with MSE reconstruction Loss.

$$\mathcal{L}_{attr}(\mathcal{G}^{aug}) = \frac{1}{|\mathcal{V}^{\mathrm{D}}|} \sum_v \mathrm{MSE}(F_v, \mathrm{MLP}(E_v))$$

# B  Dataset Statistics

Table 3: Dataset statistics

| Dataset | # Nodes | # Edges | # Classes |
|---|---|---|---|
| MAG240M | 122M | 1.3B | 153 |
| Wiki | 4.8M | 5.9M | 639 |
| arXiv | 169K | 1.2M | 40 |
| ConceptNet | 791K | 2.5M | 14 |
| FB15K-237 | 15K | 268K | 200 |
| NELL | 69K | 181K | 291 |

# C  Task Graph GNN Architecture

For the GNN over task graph $M_{\mathrm{T}}$, we use an attention-based GNN, where each node performs attention to other nodes at each layer:

$$\beta_{ij} = MLP\left(W_q^T H_i^l || W_k^T H_j^l || e_{ij}\right) \tag{18}$$

$$\alpha_{ij} = \frac{\exp(\beta_{ij})}{\sum_{k \in \mathcal{N}(i) \cup \{i\}} \exp(\beta_{ik})} \tag{19}$$

$$H_i^{l+1} = ReLU\left(BN\left(H_i^l + W_o^T \sum_{j \in \mathcal{N}(i) \cup \{i\}} \alpha_{ij} W_v^T H_j^l\right)\right) \tag{20}$$

# D  Hyperparameters

## D.1  Model Architecture, MAG240M and arxiv

We initialize the node features in citation network datasets using a pretrained language model (RoBERTa [12] base model trained on NLI and STSB). The architecture of our PromptGraph model in all of our proposed methods for citation network datasets (full PRODIGY, PG-NM, and PG-MT) and the baseline (NoPretrain), consists of two message passing layers, $M_{\mathrm{D}}$, over the DataGraph and one message passing layer, $M_{\mathrm{T}}$, over the TaskGraph. These layers are defined in Section 3.1.

For the Contrastive method, the architecture includes two message passing layers, $M_d$, over the DataGraph, and a contrastive learning component that is defined in Section 4.1. The mode for Finetune is the same as the Contrastive method, with the addition of a linear layer head over the output of the two $M_d$ layers, also described in Section 4.1.

## D.2  Model Architecture, knowledge graph datasets

We initialize node and edge features in knowledge graph datasets using a pretrained language model (MPNet [15]). The architecture of our PromptGraph model in all of our proposed methods for knowledge graph datasets (full PRODIGY, PG-NM, and PG-MT) and the baseline (NoPretrain), consists of two message passing layers, $M_{\mathrm{D}}$, over the DataGraph, an aggregator as described by

Table 4: Ablation of PG-NM on arXiv.

| Ways | PG-NM | 3 →1 shot | No Attr | No Aug | No Attr, Aug | No Attr, Aug, $M_\text{T}$ |
|------|-------|-----------|---------|--------|--------------|----------------------------|
| 3 | $\mathbf{72.50} \pm_{0.35}$ | $69.13 \pm_{1.09}$ | $65.74 \pm_{1.12}$ | $68.98 \pm_{1.09}$ | $66.53 \pm_{1.12}$ | $63.60 \pm_{1.06}$ |
| 5 | $\mathbf{61.21} \pm_{0.29}$ | $57.49 \pm_{0.92}$ | $52.78 \pm_{0.90}$ | $57.50 \pm_{0.85}$ | $53.89 \pm_{0.92}$ | $51.27 \pm_{0.69}$ |
| 10 | $\mathbf{46.12} \pm_{0.19}$ | $42.03 \pm_{0.60}$ | $37.99 \pm_{0.63}$ | $42.43 \pm_{0.64}$ | $38.87 \pm_{0.59}$ | $37.62 \pm_{0.34}$ |
| 20 | $\mathbf{33.71} \pm_{0.11}$ | $30.18 \pm_{0.38}$ | $26.60 \pm_{0.36}$ | $30.89 \pm_{0.38}$ | $27.50 \pm_{0.36}$ | $27.44 \pm_{0.17}$ |
| 40 | $\mathbf{23.69} \pm_{0.07}$ | $21.44 \pm_{0.22}$ | $18.03 \pm_{0.21}$ | $21.97 \pm_{0.24}$ | $18.52 \pm_{0.22}$ | $19.69 \pm_{0.08}$ |

Equation 3, and two message passing layers, $M_\text{T}$, over the TaskGraph, which only pass messages along the positive and query edges. These layers are defined in Section 3.1.

For the Contrastive method, the architecture includes two message passing layers, $M_d$, over the DataGraph, an aggregator as described by Equation 3 and a contrastive learning component that is defined in Section 4.1. The mode for Finetune is the same as the Contrastive method, with the addition of a linear layer head over the output of the two $M_d$ layers, also described in Section 4.1.

### D.3    Training, MAG240M

The following describes our pretraining setup:

The pretraining task we used consisted of 30 ways, 3 shots, and 4 queries per task. This specific task configuration was carefully selected to strike a balance between complexity and diversity in the training data, without overwhelming the GPU memory.

We checkpoint the model every 500 steps.

Our pretraining setup included a model with an input dimension of 768 and an embedding dimension of 256, batch size of 1, and the AdamW optimizer with a learning rate of $1 \times 10^{-3}$ and weight decay of $1 \times 10^{-3}$, a pretraining task with 30 ways, 3 shots, and 4 queries per task, and checkpointing every 500 steps. This consistent configuration was applied across all the methods for fair comparison. Our full PRODIGY setup, on average, involves sampling 1 Neighbor Matching task per 1 multitask pretraining tasks.

For our evaluation process, we computed zero-shot transfer performance of the model on the test set, using the pretraining checkpoint at the 10,000 step of pretraining. The evaluation was conducted on 500 test tasks, with batch size of 5, measured on the downstream task of graph classification accuracy. To maintain consistency, we kept the number of shots and queries constant at 3 for all evaluation tasks.

### D.4    Training, Wiki

The following describes our pretraining setup:

Our pretraining setup included a model with an input dimension of 768 and an embedding dimension of 256, the AdamW optimizer with a learning rate of $1 \times 10^{-3}$ and weight decay of $1 \times 10^{-3}$, a pretraining task with 30 ways, 3 shots, and 4 queries per task, using a batch size of 10, and checkpointing every 500 steps. This specific task configuration was carefully selected to strike a balance between complexity and diversity in the training data, without overwhelming the GPU memory. This consistent configuration was applied across all the methods for fair comparison. Our full PRODIGY setup involves sampling one neighbor matching task per 50 multitask pretraining tasks.

For our evaluation process, we computed zero-shot transfer performance of the model on the test set, using the pretraining checkpoint at the 8,000 step of pretraining. The evaluation was conducted on 500 test tasks, with batch size of 1, measured on the downstream task of graph classification accuracy. To maintain consistency, we kept the number of shots and queries constant at 3 for all evaluation tasks. We sample 1-hop neighbourhoods for ConceptNet, FB15K-237 and NELL and 2-hop neigbourhoods for Wiki.

## E    Ablation on Table 4 for the PG-NM setting

In Table 4, we ablate on various configurations of the self-supervised objective PG-NM. As described in Section 3.2, PG-NM is composed of an attribute prediction loss, dropnode and zeronode augmentations, with the default setting of sampling 3 shots neighbor matching tasks. Our best setting, referred to

as simply "PG-NM", is also shown in Table 1 and comprises of attribute prediction, dropnode and zeronode augmentations, with the default setting of 3 shots.

The ablation results reveal that using all of these elements together results in the highest performance. Specifically, attribute prediction has the greatest impact on PG-NM's performance, as its removal results in an average 7% drop across all ways, as shown in the 'No-Attr' column.

Removing the dropnode and zeronode augmentations results in an average 3% drop across all ways, as shown in the No Aug' column. Removing both attribute prediction and augmentations results in performance that is similar to just removing attribute prediction alone, which is also roughly a 7% drop across all ways, as shown in the 'No Attr, Aug' column. Additionally, we found that decreasing the number of shots to 1 from the default setting of 3 resulted in an average 3.5% drop across all ways, as shown.

## F    Evaluation using different numbers of shots

We show evaluation using different numbers of shots, as shown in Figures 3, 5, and 6, 7.

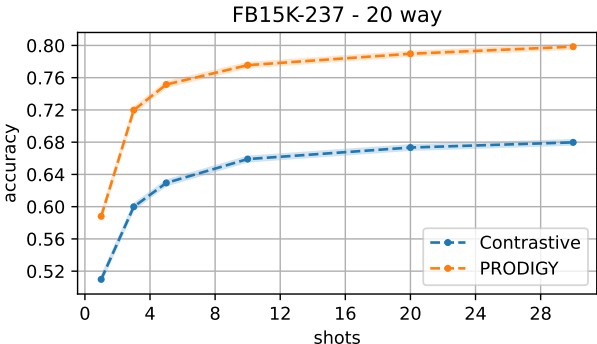

Figure 5: In-context learning accuracy on FB15K-237 in a 20-ways setting wrt. the number of prompt examples (shots).

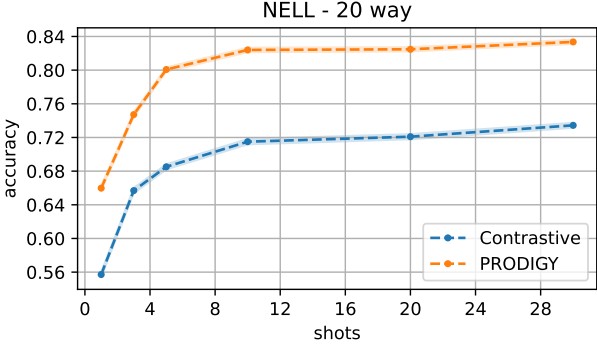

Figure 6: In-context learning accuracy on NELL in a 20-ways setting wrt. the number of prompt examples (shots).

## G    Scaling with Data Size

We explore how the model scales with more pretraining tasks. Note that we use the number of train steps as a proxy because the model sees more pretraining tasks as the training proceeds with almost no redundancy (0.20% for 10k tasks). The result on arXiv in a 5-ways setting is illustrated in Figure 4. It shows that the Contrastive baseline saturates quickly and its performance fluctuates given more

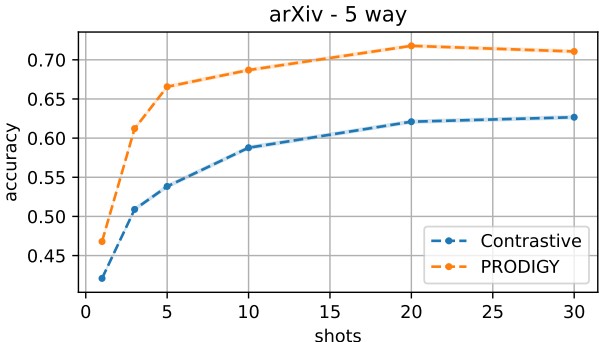

Figure 7: In-context learning accuracy on arXiv in a 5-ways setting wrt. the number of prompt examples (shots).

pretraining tasks. Instead, PRODIGY consistently shows an improvement in performance as more data is pretrained on.

# H Compute

We use one NVIDIA A100-SXM4-80GB GPU for all our experiments. One pretrain run of 10k steps takes 3 to 4 hours.

# I Broader Impacts

Our work aims to extend the success of in-context learning to graphs and start building toward graph foundation models. This would allow cost-effective and accurate predictions, especially in domains where labeled data is scarce and long tail such as network anomaly detection, rare disease diagnosis/treatment, supply chain disruption, and recommendations for new users. However, overreliance on prior knowledge from pretraining could also lead to increased social bias and unfair benefits to the dominate groups. To mitigate this, pretraining data should be diverse and well-balanced, and the pretrained models should be tested on downstream tasks over different groups and subdistributions.

