# OpenReview forum: "PRODIGY: Enabling In-context Learning Over Graphs"
_NeurIPS.cc/2023/Conference — NeurIPS 2023 spotlight_

### Official Review · Reviewer_GRdC · 2023-07-04

**Soundness:** 3 good
**Presentation:** 3 good
**Contribution:** 3 good
**Rating:** 6
**Confidence:** 3

**Summary:**

This paper proposes a pretraining framework that enables in-context learning on graph classification tasks (maybe diverse graph machine learning tasks). Specifically, it proposes a prompt graph as a unified representation for diverse tasks, then it designs a graph neural network architecture over the prompt graph and a corresponding family of in-context pre-training objectives. The experiments show that the pretrained model using this framework exhibits good in-context learning performance over various new tasks in the same domain of pre-training without finetune.


**Strengths:**

1. This paper explores a problem of great current interest – “how to enable in-context learning for diverse graph machine learning tasks”.
2. The approach is novel and technically sound.  Interestingly, the "Task graph Message Passing" step can adjust the representation of the label nodes using the prompt examples and propagate label information back to the examples and query graph representation for test-time tasks. I think this step allows prompt to provide much stronger constraint on generation than natural language prompting.
3. The paper is well-structured; The approach is clearly presented with descriptive figures.


**Weaknesses:**

1. The performance of the approach may depend on the similarity between the test and pre-training data, and it is not clear how well it transfers to different datasets.
2. The fine-tuning comparison may be insufficient, as the work does not explicitly evaluate the disparity between in-context learning and fine-tuning settings for the proposed approach.
3. The performance improvement of this work over the baseline could be partially attributed to the larger model size.


**Questions:**

1. The paper demonstrates the effectiveness of the proposed approach using the evaluation datasets from the same domain with the pre-training dataset. Does the success of this approach hinge solely on the close similarity between the test and pre-trained data? As I am not particularly conversant with the specific evaluation dataset utilized in this study, I am wondering how well is the transfer capability of this approach.
2. The Finetune baseline, sharing the same mode as the Contrastive, appears to be weak, as the observed improvement over the Contrastive seems marginal according to the experimental results. I believe a more compelling comparison would be to evaluate the performance disparity between this work within the in-context learning setting versus the finetuning setting.
3. I have concerns that the performance enhancement of this work over the baseline Contrastive may be attributed to the larger number of parameters. This work's architecture includes two message passing layers, MD and two message passing layers, MT, while the Contrastive only incorporates two message passing layers, Md. As shown in Table 4, it seems that the Contrastive model reaches saturation swiftly due to its limited capacity, while PRODIGY can persist in learning due to its greater capacity.


**Limitations:**

One limitation is that the work currently sits only on the graph classification task, and it would be nice to include some exploration of the generation task, but I won't fault the authors for that.

---

> ### Author Rebuttal · Authors · 2023-08-10
>
> We thank the reviewer for the time and valuable feedback. Below, we clarify a number of important points raised by the reviewer.
>
> > Re: Does the success of this approach hinge solely on the close similarity between the test and pre-trained data?
>
> The reviewer has concerns on whether the success of our proposed framework requires close similarity between the test and pre-trained dataset. We respectfully disagree, the pre-trained data and test data are from quite diverse domains, e.g., we aim to pretrain on wikipedia knowledge graph and perform tasks on ConceptNet and NELL, both of which are drastically different from WikiKG from both node/edge feature and graph structure point of view.
>
> > Re: The Finetune baseline, sharing the same mode as the Contrastive, appears to be weak
>
> The reviewer wonders if we adopt a weak baseline for finetune. We argue that we design our finetune baseline with contrastive learning since this is the existing best way. If we design the finetune method with our proposed method, then this is more of a variant of our own method instead of a baseline. The goal of this work is to show that with our PRODIGY framework, we for the first time allow for in-context learning over graph tasks across diverse domains, getting rid of the need of additional finetuning steps, and even outperforms traditional finetuning methods that require additional backpropagation over data on downstream domains.
>
> > Re: Number of parameters
>
> The number of parameters for contrastive learning is ~1.8m, for our method is ~2m. We acknowledge ours has a larger number of parameters but as demonstrated in the number difference, the number of parameters of MT only accounts for a small portion of the overall model parameters. We respectfully do not think this is the main reason that leads to the performance increase.

---

### Official Review · Reviewer_FSU3 · 2023-07-06

**Soundness:** 2 fair
**Presentation:** 3 good
**Contribution:** 3 good
**Rating:** 6
**Confidence:** 4

**Summary:**


This paper introduces Prodigy, a method aimed at facilitating 'in-context learning' over graphs.

The key contribution of this work is the formulation of a 'prompt' that can be utilized for in-context learning with graphs. This 'prompt' is defined as a data graph which incorporates typical (input, output) examples, akin to those used in few-shot prompting setups. Given that each node in the graph serves as an input to the method, it is crucial to contextualize each node. For this, the authors add a neighborhood to each node in the data graph. This data graph is then paired with a task graph. The task graph, designed to link different parts of the prompt (e.g., connecting nodes in the data graph that are part of the same class), comprises one node for each node in the data graph (referred to as 'data nodes'), and one node for each label (termed 'label nodes').



When running inference on a new example, the procedure begins with a Graph Convolutional Network (GCN) style message passing over the data graph, followed by the task graph. Eventually, the label is predicted based on the similarity between the representations of the query and label nodes in the task graph.

To train this model, the authors propose two self-supervised pre training objectives: neighborhood prediction and a combination of link-prediction and node prediction.

The experimental results on citation graphs and commonsense graphs indicate the potential of the Prodigy method, with it outperforming strong baselines, including fine-tuning.


**Strengths:**

The concept of extending few-shot learning to graphs is both intuitive and attractive. Based solely on the coherent design for graph-based few-shot learning, the paper is worth considering for acceptance.

**Weaknesses:**

The title's use of the term “in-context learning” is questionable. While the term is currently popular due to the rise of Large Language Models (LLMs), it may inadvertently mislead readers about the actual contributions of this paper.

Traditional in-context learning:

* Applies to a large number of tasks

* Adapts to new domains

* Allows fluid task definition

However, none of these attributes seem to apply to the proposed model. Why call it in-context learning when it essentially seems to be K-shot link prediction and node classification?

There's a crucial factor to consider: few-shot learning enables tasks to be undertaken even when no training data is available. However, PRODIGY appears to necessitate pre-existing data to operate effectively as the experiments show. This mismatch in the implementation and title is the main reason for my somewhat low score. In general, I would strongly recommend a change in title for this work (at the minimum, changing _Enabling_ to _Towards_).


**Questions:**

Q1: *L264: Then, we construct a k-shot prompt for 264 test nodes (or edges) from the test split by randomly selecting k examples per way from these available 265 examples. This allows us to test the model’s ability to learn in-context relationships and perform well on 266 classification tasks with truly limited known labels. By default we use k= 3 shots in our experiments.*

Generally, in a few-shot scenario, a fixed number of examples are included in the prompt, with the test example added at the end and supplied to the model for inference. However, the description here, which refers to the selection of both training and test examples, seems somewhat unclear. Could you please provide more clarity?


Q2: Considering the baselines, is it the most effective approach to train a single Multi-Layer Perceptron (MLP) on top of a graph encoder? Wouldn't training the entire graph encoder end-to-end yield more effective results?


**Limitations:**

Please see weaknesses.

---

> ### Author Rebuttal · Authors · 2023-08-10
>
> We thank the reviewer for the time and valuable feedback. Below, we clarify a number of important points raised by the reviewer.
>
> > Re: Difference from K-shot prediction.
>
> The reviewer raises a concern on our difference from K-shot prediction. As in the related work section, most existing few-shot learning works are designed and tested for generalizing across different tasks on the same graph. They are shown to exhibit optimal performance only when trained on similar curated tasks. Our major contribution is to relax such constraints, and enable GNNs to perform both node-level and link-level tasks across drastically different domains (citation, wikipedia, freebase, commonsense) for train/test without the need to perform additional finetuning. Overall we think Prodigy is a solid contribution to explore how we can possibly formulate and also learn in context for various graph tasks across domains. We are happy to update the title to better reflect our contribution in our final version.
>
> > Re: Generally, in a few-shot scenario, a fixed number of examples are included in the prompt, with the test example added at the end and supplied to the model for inference. However, the description here, which refers to the selection of both training and test examples, seems somewhat unclear.
>
> Yes, your understanding is correct. Given a test datapoint, we sample k training datapoints for each of the m classes/ways. Together these serve as our prompt. Note there is no test datapoint in the prompt. We will rephrase the sentence in our final version to make it easier to understand.
>
> > Re: is it the most effective approach to train a single Multi-Layer Perceptron (MLP) on top of a graph encoder? Wouldn't training the entire graph encoder end-to-end yield more effective results?
>
> The reviewer wonders if doing end-to-end training can yield better results. We have run both configurations, and we did not notice a drastic difference in performance.

---

### Official Review · Reviewer_DFam · 2023-07-07

**Soundness:** 2 fair
**Presentation:** 2 fair
**Contribution:** 2 fair
**Rating:** 6
**Confidence:** 2

**Summary:**

I have read the author’s rebuttal, I think I misunderstood the in-context learning mentioned in this paper and see the difference from other typical ICL works. I have no object to accept the paper if AC thinks it is enough contribution.

The paper introduces an in-context few-shot prompting approach for edge classification over graphs using the PRODIGY framework. The idea is to use GNN to create few shot prompts that can make LLM do better in context learning. The pretrained model exhibits strong in-context learning performance on downstream tasks, surpassing contrastive pretraining baselines and standard finetuning methods.

**Strengths:**

Originality: The paper demonstrates some novelty by utilizing a graph to generate few-shot prompting examples for classification.
Quality: The results presented show significant improvements over the baseline methods.
Clarity: The methodology is described clearly, providing a clear understanding of the different components involved in task construction.
Significance: The paper appears to lack a truly novel contribution, instead combining multiple existing approaches and claiming improvement by combining these strategies. E.g. using graphs to construct a few shot examples. How does this differ from superICL which is a more generic model that uses different downstream models for constructing in-context examples?

**Weaknesses:**

The paper is not very easy to read with some improvement room for the writing and fluency. For example, there are some incomplete sentences in the writing.  "give music product 29 recommendations on spotify when being trained on Amazon book purchasing graph."

The paper appears to lack a truly novel contribution, instead combining multiple existing approaches and claiming improvement by combining these strategies. E.g. using graphs to construct a few shot examples. How does this differ from superICL which is a more generic model that uses different downstream models for constructing in-context examples?

**Questions:**

See weakness.

---

> ### Author Rebuttal · Authors · 2023-08-10
>
> We thank the reviewer for the time and valuable feedback. Below, we clarify a number of important points raised by the reviewer.
>
> > Re: Novelty
>
> The reviewer claims that the paper lacks a truly novel contribution. We respectfully disagree. The paper proposed one of the first frameworks that allow for in-context learning over graph tasks. We are not using graphs to construct few-shot examples, but rather propose a way to unify the formulation of different types of graph tasks, including node classification and link prediction, such that the model is able to learn in context.
>
> > Re: how does this differ from superICL?
>
> We would appreciate it if the reviewer can provide a detailed reference to the paper instead of only giving a name. To the best of our knowledge, we find this paper [1] by looking up the model’s name superICL. However, we fail to find connections between our work (which focuses on enabling in-context learning for graph tasks) and superICL [1] (which leverages a combination of an LLM with smaller models to perform supervised tasks efficiently for in-context learning on text tasks). Please kindly let us know how these two are related.
>
> We will improve the writing and polish the narrative in the final version.
>
> [1] Xu, Canwen, et al. "Small models are valuable plug-ins for large language models." arXiv preprint arXiv:2305.08848 (2023).

---

### Official Review · Reviewer_GCaE · 2023-07-07

**Soundness:** 4 excellent
**Presentation:** 4 excellent
**Contribution:** 4 excellent
**Rating:** 8
**Confidence:** 4

**Summary:**

This paper proposed a framework for graph in-context learning. The PRODIGY architecture consists of the prompt graph, task graph, and in-context learning pretraining objective. The PRODIGY can directly conduct downstream tasks without finetuning and shows strong performance on downstream classification tasks.

**Strengths:**

1. The paper is overall well-organized and well-written.
2. To the best of my knowledge, this is the first work on graph in-context learning, which can be directly applied to downstream tasks without tuning.
3. The proposed PRODIGY shows strong performance, significantly outperforming baseline methods.

**Weaknesses:**

1. missing important self-supervised graph learning baselines, such as GraphMAE [1].
2. Since there is much recent progress in graph contrastive learning, including data augmentation and architecture design, I strongly encourage the authors to select more recent and competitive baselines.
3. The authors only report results of node classification on one dataset, i.e., arxiv dataset.

[1] Hou, Z., Liu, X., Cen, Y., Dong, Y., Yang, H., Wang, C., & Tang, J. (2022, August). Graphmae: Self-supervised masked graph autoencoders. In Proceedings of the 28th ACM SIGKDD Conference on Knowledge Discovery and Data Mining (pp. 594-604).


**Questions:**

The proposed PRODIGY requires pretraining and testing on the same type of graph, such as MAG240M for pretraining and arxiv for testing, where both datasets are citation networks. It would be intriguing to observe PRODIGY's performance when the training and testing data belong to different domains.

---

> ### Author Rebuttal · Authors · 2023-08-10
>
> We thank the reviewer for the time and valuable feedback. Below, we clarify a number of important points raised by the reviewer.
>
> > Re: contrastive learning baselines
>
> Thank you for the reference. The reviewer suggests we compare our method with more contrastive learning methods. Here we would like to emphasize that our main contribution is to propose the first pretraining framework that enables in-context learning over graphs. We are not simply designing a new contrastive learning algorithm, but rather the framework we proposed enables a given contrastive learning algorithm to learn in context for graph tasks. Hence we view contrastive learning algorithms to be orthogonal contributions from ours. We will implement more contrastive learning methods and do a thorough evaluation.
>
> > Re: training and test data from different domains
>
> Thank you for the suggestion. Our experiments on link prediction have demonstrated transfer across datasets from multiple domains. Enabling pre-training and testing from different domains is very challenging and requires modeling capacity to capture the gap between both node/edge features as well as the graph structure. This is exactly our future work for a more fundamental graph foundation model that can transfer and complete tasks in-context from diverse domains and even diverse tasks (node/link/graph-level prediction tasks).

---

### Decision · Program_Chairs · 2023-09-21

**Decision:**

Accept (spotlight)

**Comment:**

This paper proposes a formulation of in-context learning adapted to graphs, as well as the corresponding pre-training method to enable it.

Reviews are unanimously positive, with reviewers acknowledging the novelty of the approach and the strength of the experimental results. Based on the positive feedback, I recommend acceptance.